# Detection of salivary citrullinated cytokeratin 13 in healthy individuals and patients with rheumatoid arthritis by proteomics analysis

**Takuya Yasuda***, **Koichiro Tahara, Tetsuji Sawada**

Department of Rheumatology, Tokyo Medical University Hospital, Tokyo, Japan

* t_yasuda@tokyo-med.ac.jp

## Abstract

The immune response to citrullinated peptides in the mucosa has been suggested to play an important role in the transition from pre-onset rheumatoid arthritis (RA) to clinically evident RA. Although there are reports indicating the presence of anti-citrullinated peptide antibodies in the saliva, few studies have reported citrullinated peptide detection in human saliva. This study aimed to identify citrullinated peptides in human saliva and discuss their clinical significance. Saliva samples were collected from 11 patients with RA and from 20 healthy individuals. Citrullinated peptides were detected using an anti-modified citrulline (AMC) antibody. Saliva from the healthy individuals was subjected to two-dimensional protein electrophoresis to isolate citrullinated peptides, which were analyzed by matrix-assisted laser desorption/ionization time-of-flight mass spectrometry and mass spectrometry by peptide mass fingerprinting. The results were corroborated by immunoprecipitation (IP)-western blotting. The signal intensities of the bands precipitated with anti-cytokeratin 13 (CK13) and AMC antibodies were quantified. The signal intensity ratio of the band produced by the AMC antibody was divided by that of the band produced by the anti-CK13 antibody to calculate the citrullinated CK13 (Cit-CK13) ratio. A citrullinated peptide band corresponding to a molecular weight of approximately 50 kDa was detected in the saliva of healthy individuals, and identified as CK13 via mass spectrometry and IP-western blotting. No significant difference was observed between the salivary Cit-CK13 ratios of patients with RA and healthy participants (p = 0.605). This is the first study to show that Cit-CK13 is present in human saliva, and that there is no significant difference between the Cit-CK13 ratios of patients with RA and healthy individuals, suggesting that salivary Cit-CK13 content and RA development may not be associated. The physiological and pathological roles of Cit-CK13 in the oral cavity, and its responsiveness to mucosal immunity, remain unknown and will be the subject of further investigation.

**Data Availability Statement:** All relevant data are within the manuscript files.

**Funding:** The authors received no specific funding for this work.

**Competing interests:** The authors have declared that no competing interests exist.

## Introduction

Rheumatoid arthritis (RA) is an autoimmune disease, in which the synovium is the primary site of the disease. Citrullinated peptides are thought to be involved in the pathogenesis of RA [1]. Fibrin (fibrinogen) [2, 3], vimentin [4], fibronectin [5], and α-enolase [6] have been reported as citrullinated peptides present in the joint synovium and joint fluid of patients with RA. Recent studies have indicated that in the pre-onset period, immune responses to citrullinated peptides in mucosal membranes, such as those present in the lungs and intestinal tract, may play an important role in the transition from pre-onset RA to clinically evident RA [7–9]. As a citrullinated peptide present in normal mucosal tissues, citrullinated filaggrin is reportedly expressed in the mucosal epithelial cells of the skin, oral cavity, esophagus, and nasal cavity. Filaggrin undergoes citrullination by peptidylarginine deiminase (PAD) during epithelial cell differentiation and plays an important role in the physiological function of epithelial cells [10, 11].

Citrullinated peptides are produced by the post-translational modification of arginine residues in proteins to citrulline residues by the action of PAD in the presence of calcium ions. Human citrullinated enzymes have five isoforms: PAD1, PAD2, PAD3, PAD4 (identical to PAD5), and PAD6, with each isoform exhibiting a different tissue localization and substrate profile. PAD2 and PAD4 are primarily involved in the pathogenesis of RA and other immune diseases [12]. PAD2 is ubiquitously expressed in several tissues, including skeletal muscles and the spleen, brain, and secretory glands, whereas PAD4 is primarily expressed in neutrophils and monocytes [13]. Although microorganisms do not normally express PAD, *Porphyromonas gingivalis* (Pg), a pathogen of periodontal disease, has been reported to produce *Porphyromonas* PAD (PPAD) and citrullinated peptides in periodontal tissues [14].

Anti-citrullinated peptide antibodies (ACPAs) have been detected in the sera of patients with RA and are widely used for RA diagnosis because of their excellent sensitivity and specificity [15]. Although there have been several reports on ACPAs in the saliva [16–20], there are limited reports on citrullinated peptides as autoantigens in saliva. This study aimed to identify previously unidentified citrullinated peptides in human saliva through a comprehensive investigation of human saliva, and to discuss their clinical significance.

## Materials and methods

### Samples

After screening based on a questionnaire developed by Sugihara et al. [21], saliva samples were collected from participants with no subjective symptoms of periodontal disease (11 patients with RA and 20 healthy participants). For saliva from healthy individuals, samples were used for experiments in the order of collection, and 10 subjects were used for experiments to detect citrullinated peptides in saliva by Western blot, and the other 10 subjects for semi-quantitative study of the citrullinated peptide identified in saliva by mass spectrometry. Written informed consent was obtained from all individuals for participation in the study. Serum was collected by centrifugation of whole blood at 3000 rpm for 10 min and stored in a -20˚C freezer until subsequent experimentation. Saliva was collected after rinsing the mouth with water and stored in a freezer at -20˚C until subsequent experimentation. This study was approved by the Medical Ethics Review Committee of Tokyo Medical University.

### Detection of citrullinated peptides

The proteins in the samples were separated by sodium dodecyl sulfate-polyacrylamide gel electrophoresis (SDS-PAGE), followed by western blotting and membrane chemical treatment.

After chemical treatment, an anti-chemically modified citrulline (anti-modified citrulline, AMC) rabbit polyclonal immunoglobulin (Ig) G antibody (ROI004, SHIMA Laboratories) was used to detect citrullinated peptides. This AMC antibody is the gold standard reagent for the detection of citrullinated peptides and is a polyclonal antibody that specifically recognizes chemically modified peptidyl-citrulline residue [22]. The detection method is briefly shown below. To chemically modify the citrulline residues of citrullinated peptides transferred onto a polyvinylidene difluoride (PVDF) membrane, the membrane was chemically modified as per the manufacturer's instructions (ROI004, SHIMA Laboratories) with the following solution: 0.0125% $FeCl_3$, 2.3 M $H_2SO_4$, 1.5 M $H_3PO_4$, 0.25% diacetyl monoxime, 0.125% antipyrine, and 0.25% acetic acid at 37°C for overnight incubation [22]. The membrane was then washed once with deionized water, blocked overnight at 4°C with Block Ace (Dainippon Pharmaceutical)/Tris-buffered saline containing 0.05% Tween (TBST), and probed with the AMC antibody diluted 1,000-fold in 10% Block Ace/TBST for 3 h at room temperature. The membrane was then washed three times with TBST and probed with a horseradish peroxidase (HRPO)-conjugated anti-rabbit IgG antibody diluted 20,000-fold in 10% Block Ace/TBST for 90 min at room temperature. After washing three times with TBST, the membrane was visualized by chemiluminescence using the ECL Prime western blotting Detection Reagent (RPN2232, Amersham). The chemiluminescence signals of the bands were detected using ChemiDoc XRSPlus (1708265, Bio-Rad). To exclude the possibility of cross-reactivity of secondary antibodies (HRPO-conjugated goat anti-rabbit IgG antibody) with IgG included in human saliva, WB was performed simultaneously under the above conditions with and without primary antibody.

Thrombomodulin (873339, Asahi Kasei Pharma) citrullinated in vitro by PAD from rabbit skeletal muscle (P1584, Sigma-Aldrich) in a reaction buffer containing 100 mM Tris (pH 7.6), 100 mM calcium chloride, and 5 mM dithiothreitol was used as a positive control for the detection of citrullinated peptides.

## Identification of citrullinated peptides in saliva samples

Using the ZOOM IPGRunner system (ThermoFisher Scientific), 15 μg of each saliva sample were applied to a carrier amphipathic electrolyte isoelectrophoresis gel and separated, and the second dimension was developed using SDS-PAGE. To isolate citrullinated peptides from saliva samples electrophoresed by two-dimensional electrophoresis, two identical two-dimensional electrophoresis gels were prepared. One gel was subjected to western blotting using the AMC antibody, and the other sheet was subjected to silver staining using Pierce Silver Stain for mass spectrometry (24600, Thermo Scientific) to detect total protein. These two gels were compared, and the part of the gel that matched the spot of citrullinated peptide in the first gel was excised, and digested with trypsin (Promega, Madison, WI) without alkylation. The sample solution was desalted using ZipTip (Millipore) and then ionized using α-cyano-4-hydroxycinnamic acid as a matrix [23], and subjected to matrix-assisted laser desorption/ionization time-of-flight mass spectrometry (MALDI-TOF MS, Microflex LRF20 [Bruker Daltonics]). Mass spectrometry was performed using the peptide mass fingerprinting (PMF) method (Genomine).

## Immunoprecipitation (IP)-western blotting

Citrullinated cytokeratin 13 (Cit-CK13) was immunoprecipitated from saliva samples. To capture the target antigen, Cit-CK13, the Dynabeads Protein G Immunoprecipitation kit (MAN0017348, Invitrogen) and anti-human CK13 polyclonal antibody (LS-C664954, Life-Span BioSciences) were used. Briefly, to promote the binding between the magnetic beads and the anti-CK13 antibody, 200 μL of an antibody binding and washing buffer and 5 μL (10 μg) of an anti-CK13 polyclonal antibody were added to a tube containing 1.5 mg of magnetic beads

and incubated for 10 min at room temperature under rotating conditions. The tube was placed on a magnet to remove the supernatant, following which 1000 μL of a vigorously vortexed saliva sample was added, and the mixture was reacted by rotation at room temperature for 10 min to ensure that the magnetic bead-anti-CK13 antibody complex bound to CK13. The samples were washed once with Ab binding and washing buffer and washed three times with a washing buffer. After removing the washing buffer, 20 μL of the elution buffer, 2.5 μL of NuPAGE LDS Sample buffer (NP0007, Thermo Fisher Scientific), 1 μL of 0.5 M dithiothreitol, and 6.5 μL of deionized water were added to the magnetic bead-antibody-antigen complex and heated at 70°C for 10 min. Following this, 12 μL of the supernatant was separated by SDS-PAGE and subjected to western blotting. Two membranes were prepared: one was chemically treated and immunostained with the AMC antibody, and the other was immunostained with the anti-CK13 antibody.

Immunostaining with the anti-CK13 antibody was performed by incubating with an anti-CK13 antibody diluted 1,000-fold in 10% Block Ace/TBST for 3 h at room temperature, washing three times with TBST, and incubating with the HRPO-conjugated EasyBlot anti-rabbit IgG antibody diluted 5,000-fold in 10% Block Ace/TBST (GTX 221666–01, GeneTex) for 90 min at room temperature. After washing three times with TBST, the membranes were visualized using chemiluminescence.

### Quantification of Cit-CK13 in saliva

Semi-quantitative analysis was performed based on the signal intensity of the bands detected using western blotting with Chemidoc XRSPlus. Two gels were prepared by SDS-PAGE using 12 μL of saliva from patients with RA and healthy individuals per lane, and western blotting was performed. In this case, the same saliva specimen was electrophoresed as a standard sample for each immunostaining experiment. One blot was immunostained with the anti-CK13 antibody, and the other blot was chemically treated and immunostained with the AMC antibody. To exclude the possibility of cross-reactivity of secondary antibodies (HRPO-conjugated goat anti-rabbit IgG antibody) with IgG included in human saliva, WB was performed simultaneously under the conditions with and without primary antibody. Next, the bands of anti-CK13 and AMC antibodies detected using Chemidoc XRSPlus were designated by lines according to the manufacturer's instructions, and the signal intensities of the bands in the designated areas were quantified. The signal intensity ratio was calculated by dividing the signal intensity of the anti-CK13 antibody band and the AMC antibody band of each sample by the signal intensity of the standard sample. The signal intensity ratio of the band and the AMC antibody was divided by the signal intensity ratio of the band and the anti-CK13 antibody to calculate the Cit-CK13 ratio.

### Statistical analysis

Statistical analyses were performed using the SPSS software (IBM, version 28). Median comparisons were performed using the Mann–Whitney test, with the Shapiro–Wilk test for normality. The results of the two-tailed test with $p < 0.05$ were considered statistically significant.

## Results

### Detection of citrullinated peptides in saliva

The participants had a median age of 32.0 years (95% confidence interval [CI]: 28.1–39.4; 10 males). S1A Table shows the age and smoking history of the 10 participants. Western blotting revealed the presence of a band of citrullinated peptide of approximately 50 kDa in all samples.

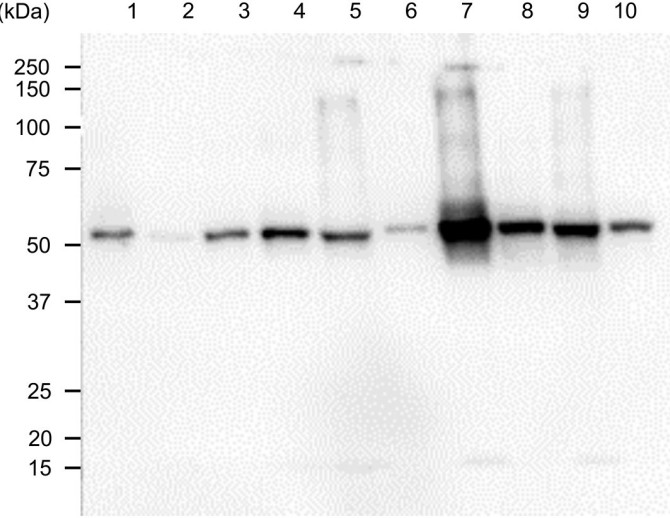

**Fig 1. Citrullinated peptides in the saliva of healthy participants.** Twelve microliters of saliva per lane were subjected to sodium dodecyl sulfate-polyacrylamide gel electrophoresis. The membrane was chemically treated and immunostained using an anti-modified citrulline antibody as the primary antibody and horseradish peroxidase-labeled goat anti-rabbit IgG antibody as the secondary antibody. Lanes 1–10:healthy human saliva.

Considering the possibility that the secondary antibodies reacted with IgGs in human saliva, we set up a lane in which only HRPO-conjugated EasyBlot goat anti-rabbit IgG antibody was reacted with healthy human saliva without using primary antibodies, but no bands were detected. Fig 1 shows the results of western blotting for the representative cases.

## Identification of candidate citrullinated peptide in saliva

Two-dimensional electrophoresis and silver staining of saliva samples from healthy participants at 200 μg per lane revealed the presence of a large number of peptides (Fig 2A). The spot corresponding to the positive spot stained with the AMC antibody (Fig 2B arrow) was excised, digested in gel with trypsin, and subjected to mass spectrometry using MALDI-TOF MS/PMF. The excised protein spot was identified as cytokeratin 13 (accession number: P13646). S2 Table shows the amino acid sequence of cytokeratin 13 and the amino acid sequence identified by MALD-TOF MS/PMF analysis (underlined).

## Salivary Cit-CK13 identified by IP-western blotting

To show that salivary CK13 was citrullinated, saliva samples from the same participants (samples used for mass spectrometry) were immunoprecipitated with an anti-CK13 antibody and then immunostained with an anti-CK13 antibody (Fig 3A) and AMC antibody (Fig 3B). As shown in Fig 3, proteins of approximately 50 kDa immunoprecipitated by the anti-CK13 antibody from saliva reacted with the anti-CK13 and AMC antibodies, which confirmed that the citrullinated peptide present in saliva was Cit-CK13.

## Quantification of Cit-CK13 in saliva

Representative western blotting images of salivary CK13 and Cit-CK13 in 11 patients with RA (age [years]: median 65.0, 95% confidence interval: 56.9–73.9; two males, nine females) and ten healthy individuals (age [years]: median 30.5, 95% confidence interval: 29.4–31.2; four

**(A)**

**(B)**

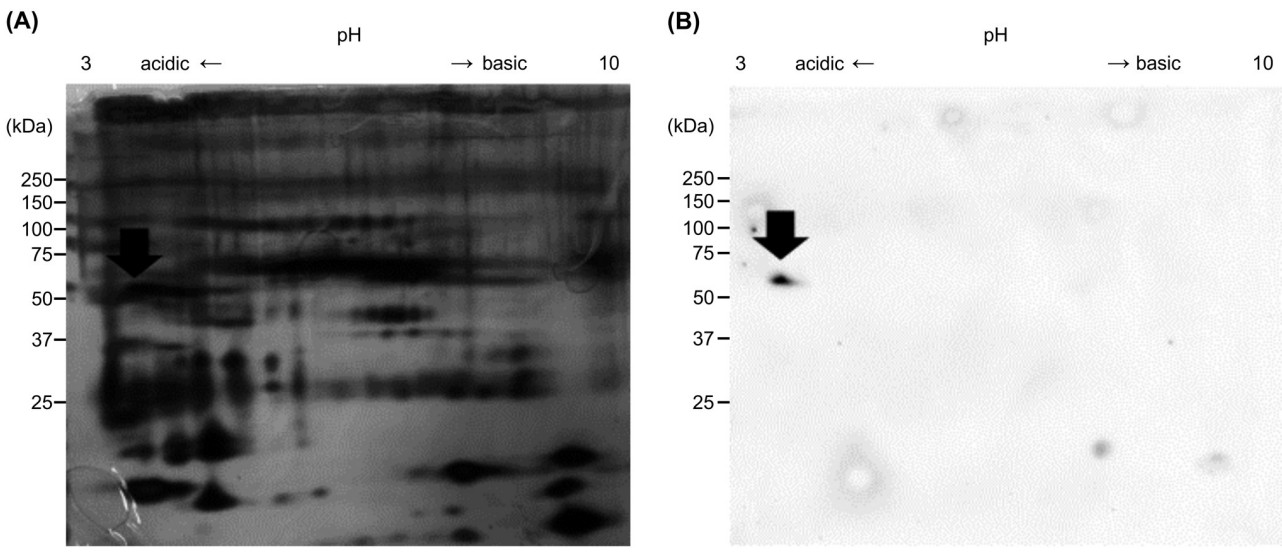

**Fig 2. Two-dimensional electrophoresis of citrullinated peptides in the saliva of healthy participants.** Saliva samples (200 μg per lane) were used. A: silver stain. B: the membranes were chemically treated and immunostained using an anti-modified citrulline antibody as the primary antibody and horseradish peroxidase-labeled goat anti-rabbit IgG antibody as the secondary antibody. Arrows: the peptide was excised and subjected to matrix-assisted laser desorption/ionization time-of-flight mass spectrometry analysis.

males, six females) are shown in Figs 4 and 5. S1B and S1C Table shows the raw data including their smoking history. As shown in Fig 4, the western blotting of saliva samples from patients with RA (lanes 1–5) and healthy individuals (lanes 6–11) using the anti-CK13 antibody as the primary antibody led to the formation of a band of approximately 50 kDa in all lanes. A band was detected in the presence of the primary antibody (Fig 4A), but not in the absence of the primary antibody (Fig 4B). Therefore, it is very unlikely that the secondary antibody binds to the saliva sample and generates a band in this WB. As shown in Fig 5, the western blotting of saliva samples from patients with RA (lanes 1–5) and healthy individuals (lanes 6–11) using the AMC antibody as the primary antibody led to the detection of a citrullinated peptide of approximately 50 kDa in all lanes. A band was detected in the presence of the primary antibody (Fig 5A), but not in the absence of the primary antibody (Fig 5B). Therefore, it is very unlikely that the secondary antibody binds to the saliva sample and generates a band in this WB.

For the western blots of CK13 and Cit-CK13, the chemiluminescence signals of the bands were measured, and the salivary Cit-CK13 ratio was calculated quantitatively. As shown in Fig 6, there was no statistically significant difference between the salivary Cit-CK13 ratios of patients with RA (n = 11) and healthy individuals (n = 10; p = 0.605).

## Discussion

We detected citrullinated peptides in saliva samples from healthy individuals using western blotting. There have been no previous reports of CK13 detection in saliva. To our knowledge, this is the first study to identify Cit-CK13 as a citrullinated peptide in saliva.

With respect to the presence of citrullinated peptides in human saliva, Tar et al. developed an assay system to immobilize saliva samples on membranes and perform the semi-quantification of total citrullinated peptides using AMC antibodies, and reported that the quantities of total citrullinated peptides in saliva from patients with RA and healthy controls were

**(A)** IP : CK13
WB: CK13

**(B)** IP : CK13
WB: AMC

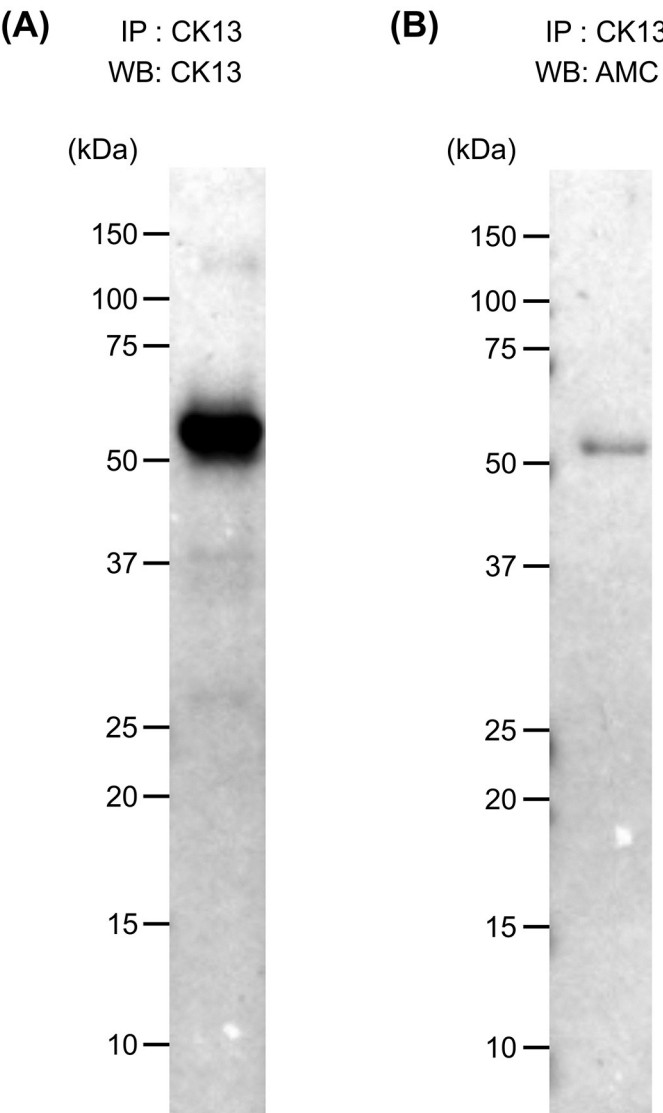

**Fig 3. Immunoprecipitation (IP)-western blotting (WB) of saliva samples from healthy participants.** A. representative IP-western blot is shown. IP: anti-cytokeratin (CK) 13 antibody; WB: anti-cytokeratin (CK) 13 antibody; B. anti-modified citrulline (AMC) antibody after chemical treatment.

comparable [24]. Because their assay system was based on a dot-blot method, the details of salivary citrullinated peptides remained unknown. In contrast, we performed a comprehensive analysis of citrullinated peptides in saliva using two-dimensional electrophoresis. Although several peptides were detected in the saliva samples by silver staining, the number of citrullinated peptides detected using AMC antibodies was unexpectedly small, and Cit-CK13 was the major salivary citrullinated peptide. Since the dot blot method developed by Tar et al. could detect citrullinated peptides present at levels below the detection sensitivity of the western blotting method used by us, the influence of the minor groups of citrullinated peptides cannot be denied. However, we found no significant difference between the levels of Cit-CK13, a major citrullinated peptide that we identified in the saliva samples, in patients with RA and healthy

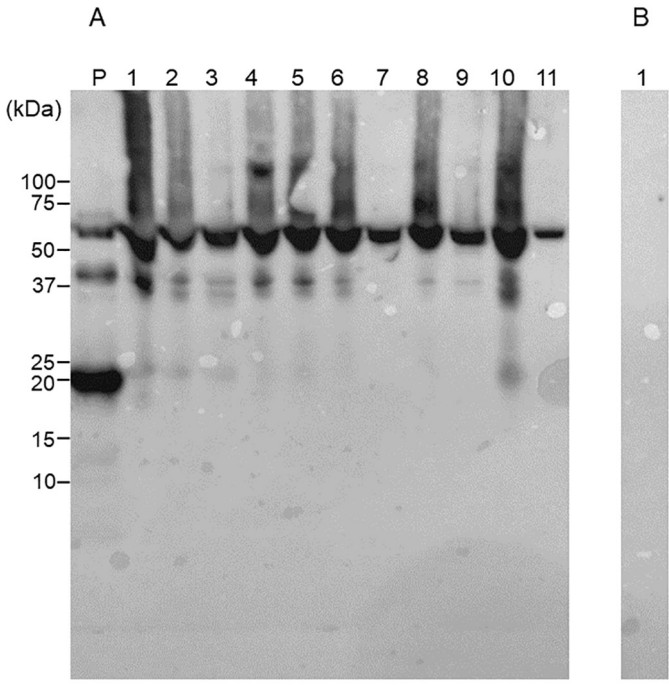

**Fig 4. Cytokeratin (CK) 13 in the saliva of patients with rheumatoid arthritis (RA) and healthy participants.**
Twelve microliters of saliva per lane were subjected to sodium dodecyl sulfate-polyacrylamide gel electrophoresis. A:
Immunostaining was performed using an anti-CK13 antibody as the primary antibody and horseradish peroxidase-
conjugated goat anti-rabbit IgG antibody as the secondary antibody. Lane P: positive control (recombinant CK13),
Lanes 1–5: saliva from patients with RA; Lanes 6–11: saliva from healthy participants. B: Immunostaining was
performed using only HRPO-conjugated goat anti-rabbit IgG antibody without primary antibody. Lane 1: Saliva from
a healthy individual.

controls Therefore, it is highly probable that the human salivary citrullinated peptide detected
by Tar et al. was Cit-CK13, although we did not perform the dot blot analysis ourselves.

We detected Cit-CK13 as a citrullinated peptide in the saliva of patients with RA and
healthy individuals that was electrophoresed approximately up to 50 kDa. Conversely, Sakagu-
chi et al. conducted an analysis using a mouse type II collagen-induced arthritis (CIA) model
to clarify the relationship between periodontal disease and arthritis and showed that Pg admin-
istration in the oral cavity of mice with CIA worsened the clinical parameters of arthritis and
histological findings such as inflammatory cell infiltration and bone joint destruction [19].
Interestingly, they also analyzed citrullinated autoantigens and reported the detection of 55
kDa citrullinated peptides in the serum, saliva, and arthritis-affected sites in the CIA group, in
the presence as well as in the absence of oral Pg administration. In contrast, no citrullinated
peptide was detected in the saliva of healthy mice that received Pg orally. Therefore, the 55
kDa citrullinated peptide is an interesting molecule in that it is not citrullinated by PPAD, but
is a salivary citrullinated peptide detected with the progression of arthritis. The authors did not
report the identity of the 55 kDa peptide. Comparable to the molecular weight of Cit-CK13,
which we identified, the molecular weight of both proteins was approximately 50 kDa, but the
expression patterns were different in the presence and absence of arthritis, suggesting that they
are likely to be different proteins. However, we cannot deny the possibility that the physiologi-
cal distribution of Cit-CK13 differs between mice and humans owing to differences between
the species. Additionally, it is not possible to conclude that the proteins differed only in their

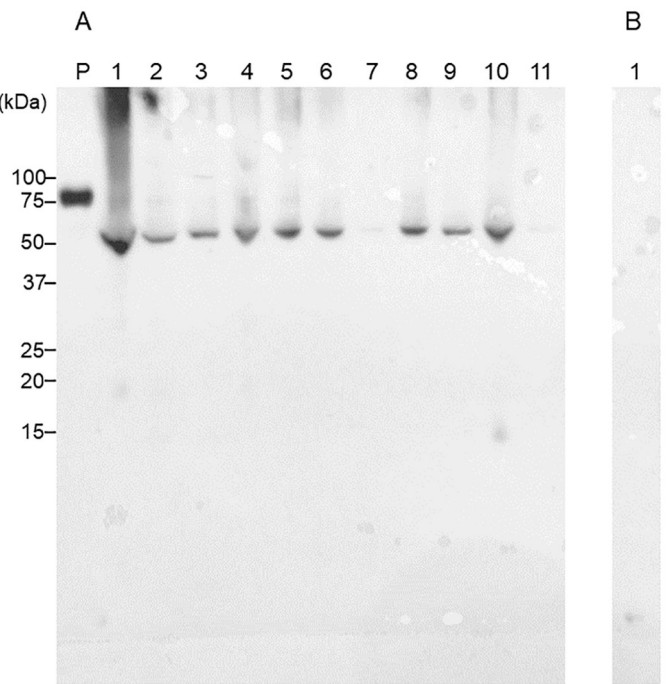

**Fig 5. Citrullinated peptides in the saliva of patients with rheumatoid arthritis (RA) and healthy participants.**
Twelve microliters of saliva per lane were subjected to sodium dodecyl sulfate-polyacrylamide gel electrophoresis. A:
Immunostaining was performed using an anti-modified citrulline antibody as the primary antibody and a horseradish
peroxidase-labeled goat anti-rabbit IgG antibody as the secondary antibody after the chemical treatment of the
membrane. Lane P: positive control (citrullinated thrombomodulin); Lanes 1–5: saliva from patients with RA; Lanes
6–11: saliva from healthy participants. B: Immunostaining was performed using only HRPO-conjugated goat anti-
rabbit IgG antibody without primary antibody. Lane 1: Saliva from a healthy individual.

expression patterns. Findings from the time-of-flight mass spectrometry analysis of the 50 kDa
protein in the saliva of mice with CIA are necessary to obtain conclusive evidence on this
topic.

The site of CK13 citrullination remains unknown. The citrullination of salivary peptides
could occur via the leakage of PAD from cells, or via the secretion or leakage of citrullinated
peptides into the saliva within salivary gland cells. We searched the Genotype-Tissue Expres-
sion (GTEx) database for PAD expression in salivary glands [25]. While the parotid and sub-
mandibular glands were not included in this database entry, minor salivary glands were
included. According to the GTEx search results, all PADs are expressed in human minor sali-
vary glands. The average calcium concentration in human saliva is 5.8 mg/dL (1.45 mM) [26],
which is suitable for maintaining the activity of PAD that is produced in the salivary gland
cells and leaks into the saliva, which is sufficient for supporting the enzymatic activity of PAD
[27]. Conversely, the cytoplasmic calcium concentration is approximately 100 nM [28], and
the intracellular calcium concentration may increase via parasympathetic stimulation [29].
Histones and other proteins have been reported to be citrullinated intracellularly [30, 31]. And
it is possible that PADs expressed in salivary glands intracellularly citrullinate CK13, resulting
in the secretion of Cit-CK13 into saliva.

With respect to the role of citrullination in the oral mucosa, Arita et al. reported that PAD
contributes to the integrity of the stratum corneum of the oral mucosa by citrullinating filag-
grin and profilaggrin in the rat palate [32]. In addition to filaggrin, other citrullinated peptides

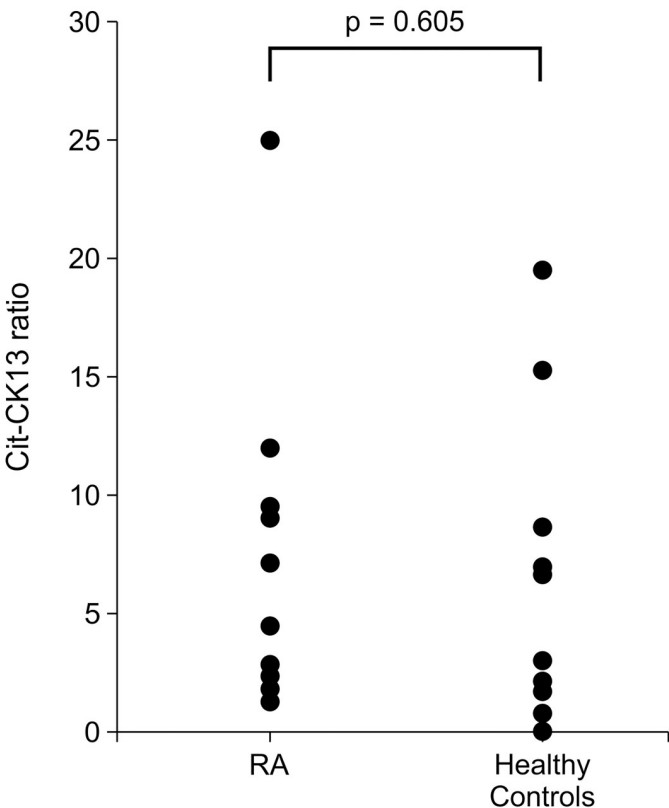

**Fig 6. Distribution of citrullinated cytokeratin 13 (Cit-CK13) ratio in the saliva of patients with rheumatoid arthritis (RA) and healthy participants.** The signal intensities of bands corresponding to cytokeratin 13 (CK13) and Cit-CK13 in saliva samples were measured. The same saliva sample was used as a standard sample for each immunostaining experiment, and the signal intensity ratio was calculated by dividing the signal intensities of the CK13 and Cit-CK13 bands of each saliva sample by the signal intensity of the standard sample. The Cit-CK13 ratio was calculated by dividing the signal intensity ratio of the Cit-CK13 band by that of the CK13 band.

with physiological roles include the glial cell fibrillary acidic protein [33], myelin basic protein [34], and histones [35]. For example, physiologically, citrullinated vimentin causes intermediate filament proteins to lose their ability to polymerize and form filaments [36], and citrullinated histones are involved in the regulation of gene expression and neutrophil extracellular trap formation. The physiological role of oral Cit-CK13 remains unknown, but similar to citrullinated filaggrin, it may play an important role in maintaining the structure and function of the oral mucosa; further investigation is needed on this topic.

A limitation of the present study is that the diagnosis of periodontal disease was not confirmed by dentists and oral surgeons. Screening for periodontal disease is generally performed by measuring the depth of the periodontal pockets, and the diagnosis of periodontal disease is confirmed when the depth is 4 mm or more [37]. In contrast to this, Sugihara et al. examined the usefulness of questionnaire-based screening in 990 Japanese individuals and reported that it was possible to efficiently screen patients with periodontal disease using a symptom-based screening questionnaire [21]. All participants in this study showed negative screening results, as assessed by the authors, and showed no subjective symptoms of periodontal disease. Furthermore, in Japan, in the age group corresponding to the age group of the healthy individuals included in this study (median age: 31.0 years), only 32.4% individuals had periodontal pockets with depths of 4 mm or more [38]. Our identification of Cit-CK13 is unlikely to be related to

periodontal disease, since it was detected in all healthy individuals. Another limitation of the study is that it only included RA as a disease control and did not include other autoimmune diseases, including Sjogren's syndrome, or neoplastic diseases such as salivary gland tumors. Citrullination has been reported to be associated not only with RA but also with the pathogenesis of malignancy [39]. Therefore, new findings may be obtained by targeting other oral diseases, including neoplastic diseases, which can be considered a subject for future research.

Although the IgG class ACPA (IgG-ACPA) content is measured in the routine diagnosis of RA, reportedly, there are IgM, and IgA classes of ACPA that are highly specific for RA as well as IgG-ACPA [40, 41]. IgG-ACPA binds to citrullinated peptides to form immune complexes and is considered to be involved in the induction and prolongation of rheumatoid synovitis [42]. IgA is an Ig that plays an important role in biological defense in mucosal tissues. IgA class ACPA (IgA-ACPA), similar to IgG-ACPA, was detected in the blood before the onset of RA [8]. Furthermore, IgA-ACPA is associated with high disease activity [18] and has been reported to be more useful than IgG-ACPA as a long-term prognostic factor for joint degeneration in early RA [43]. Therefore, IgA-ACPA may also play a facilitative role in RA pathogenesis. However, it was shown that IgA-ACPA may play a protective role in idiopathic pulmonary fibrosis (IPF) [44]. In vitro and in vivo experiments have shown that citrullinated vimentin promotes fibrosis in IPF [45]. Interestingly, Matson et al. recently reported that ACPA was detected in bronchoalveolar lavage fluid in approximately 20% of patients with IPF, and the prognosis of patients with IPF who tested positive for IgA-ACPA was significantly better than that of patients with IPF who tested negative for ACPA. This suggests that IgA-ACPA may exert a protective effect in the fibrotic process in IPF [44]. IgA-ACPA has been reported to be present not only in blood but also in saliva [18]. The physiological and pathological roles of Cit-CK13 in the oral cavity and its reactivity with IgA-ACPA are unknown and may be investigated in future studies.

## Conclusion

Herein, we demonstrated for the first time the presence of Cit-CK13 in human saliva through a comprehensive assay for salivary citrullinated peptides. We measured the ratio of oral CK13 to Cit-CK13, considering that a difference in the ratio between the Cit-CK13 levels of patients with RA and healthy participants would provide insight into the pathogenesis of RA and the mechanism underlying ACPA production in the pre-RA stage. Although the Cit-CK13 ratio is assumed to correspond to the ratio of salivary Cit-CK13, the present analysis of the Cit-CK13 ratio showed no significant difference between the contents in patients with RA and healthy participants, suggesting that CK13 citrullination does not affect RA development. It remains to be confirmed in future studies whether Cit-CK13 influences the structure and function of the oral mucosa in the oral cavity.

## Supporting information

**S1 Table. Characteristics of healthy subjects (A, B) and rheumatoid arthritis (RA) patients (C) whose saliva were collected.**
(DOCX)

**S2 Table. Amino acid sequences of cytokeratin13 peptide identified by matrix 2 assisted laser desorption/ionization time of flight (MALDI-TOF) mass spectrometry.**
(DOCX)

**S1 Raw images.**
(PDF)

## Author Contributions

**Conceptualization:** Koichiro Tahara, Tetsuji Sawada.

**Data curation:** Takuya Yasuda.

**Formal analysis:** Takuya Yasuda.

**Investigation:** Takuya Yasuda.

**Methodology:** Takuya Yasuda, Koichiro Tahara, Tetsuji Sawada.

**Project administration:** Takuya Yasuda, Koichiro Tahara, Tetsuji Sawada.

**Resources:** Takuya Yasuda, Koichiro Tahara.

**Supervision:** Koichiro Tahara, Tetsuji Sawada.

**Validation:** Tetsuji Sawada.

**Visualization:** Takuya Yasuda.

**Writing – original draft:** Takuya Yasuda.

**Writing – review & editing:** Takuya Yasuda, Tetsuji Sawada.

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
