## [Decision Letter · Decision Letter 0]

14 Jan 2022

PONE-D-21-37361Detection of salivary citrullinated cytokeratin 13 in healthy individuals and patients with rheumatoid arthritis by proteomics analysisPLOS ONE

Dear Dr. Takuya Yasuda,

Thank you for submitting your manuscript to PLOS ONE. After careful consideration, we feel that it has merit but does not fully meet PLOS ONE’s publication criteria as it currently stands. Therefore, we invite you to submit a revised version of the manuscript that addresses the points raised during the review process.

We look forward to receiving your revised manuscript.

Kind regards,

Oksana Lockridge, Ph.D.

Academic Editor

PLOS ONE

Journal Requirements:

Reviewers' comments:

Reviewer's Responses to Questions

**Comments to the Author**

1. Is the manuscript technically sound, and do the data support the conclusions?

Reviewer #1: Yes

Reviewer #2: Yes

2. Has the statistical analysis been performed appropriately and rigorously? 

Reviewer #1: Yes

Reviewer #2: Yes

3. Have the authors made all data underlying the findings in their manuscript fully available?

Reviewer #1: Yes

Reviewer #2: Yes

4. Is the manuscript presented in an intelligible fashion and written in standard English?

Reviewer #1: Yes

Reviewer #2: Yes

5. Review Comments to the Author

Reviewer #1: Summary: Citrullinated proteins in the joint synovium and joint fluid are diagnostic for rheumatoid arthritis. The present report tested the possibility that citrullinated proteins in human saliva could be used for diagnosis of rheumatoid arthritis. The first goal was to determine if saliva contained citrullinated proteins. Western blotting showed that human saliva contains citrullinated protein with a molecular weight of about 50 kDa. The citrullinated protein was identified as cytokeratin 13. The procedure for identifying the citrullinated protein included isolation by 2D gel electrophoresis, trypsin digestion, MALDI-TOF mass spectrometry and assignment of tryptic peptides to a protein. Comparison of band intensities on Western blots for citrullinated cytokeratin 13 in the saliva of healthy persons and rheumatoid arthritis patients showed no statistical differences. It was concluded that salivary citrullinated cytokeratin 13 levels are not associated with rheumatoid arthritis.

1. Page 4 line 92. Please give a brief description of the chemical treatment and why it is necessary to perform the chemical treatment on the PVDF membrane. Perhaps the target protein binds to PVDF membrane only if the membrane is chemically treated. Perhaps the antibody binds to citrullinated protein only if the membrane is chemically treated. It is not clear whether the chemical treatment affects the PVDF membrane or proteins on the membrane.

2. Page 7 line 152. It seems the chemical treatment was performed on the blot after protein had been transferred from the gel to the PVDF membrane. Does chemical treatment convert arginine to citrulline?

3. Page 4 line 92. Please name the manufacturer whose instructions you followed for chemically treating the PVDF membrane.

4. Page 5 line 117. Please include the name of the manufacturer and the model of the MALDI-TOF mass spectrometer.

5. Page 5 line 118. Peptide mass fingerprinting requires the protein to be digested with trypsin. Please describe how the sample was prepared for MALDI-TOF mass spectrometry. Was the protein reduced and alkylated? Were the peptides desalted before application to the target plate? What matrix was applied to the sample to make it ionize in the mass spectrometer?

6. page 7 lines 151-153. Typing error “one gel was immunostained with the anti-CK13 antibody, and the other gel was chemically treated and immunostained with the AMC antibody.” The typing error is the word “gel”. The word gel should be blot.

7. Page 8 line 189. The UniProt accession number for human cytokeratin 13 protein is P13646. The K1C13 name is the name of the gene.

8. Page 9 line 207. Typing error FigIP should be IP

9. Please provide the amino acid sequences of cytokeratin 13 peptides identified by MALDI-TOF mass spectrometry.

Reviewer #2: I am happy to take an opportunity to review the manuscript entitled “Detection of salivary citrullinated cytokeratin 13 in healthy individuals and patients with rheumatoid arthritis by proteomics analysis”. The study focused on the citrullinated protein in human saliva and compared that in healthy subjects with patients with rheumatoid arthritis. This topic is important in exploring the etiology of rheumatoid arthritis. This manuscript is a novel work in that it shows citrullinated CK13 is identified in saliva. However, some points as indicated below need to be addressed by authors to improve the quality of the article.

Specific comments are listed below.

1. The authors should explain why CK13 was focused on as a protein in saliva.

2. It is unclear how serum collected from RA and healthy individuals were used in this study.

3. The patients’ characteristics, including smoking history and serological profile of healthy subjects and RA patients, should be appropriately described.

4. The authors should explain how 10 patients to be compared with RA patients were selected from 21 healthy controls.

5. “p = 0.705” in the 242th line should be corrected to “p = 0.605” as shown in Fig 6.

6. PLOS authors have the option to publish the peer review history of their article (what does this mean?). If published, this will include your full peer review and any attached files.

Reviewer #1: No

Reviewer #2: No

---

## [Author Response · Author response to Decision Letter 0]

28 Feb 2022

1-4 Reviewer Comments to the Author

We appreciate the reviewers' assessment of our manuscript.

To the Editor, Reviewer 1, and Reviewer 2

Although this is not a response to the points raised, an error was found in the proofreading process. Therefore, we would like to correct the following one point.

Page 8 line 190.

(Before correction): we set up a lane in which only HRPO-conjugated EasyBlot goat anti-rabbit IgG antibody was reacted with healthy human saliva without using primary antibodies 

(After　correction): "EasyBlot" was deleted.

(Reason) It was a simple mistake. The EasyBlot secondary antibody was used in the experiment "Immunoprecipitation-Western blotting". However, EasyBlot was not used for detection of citrullinated peptides in saliva. 

In addition, we have made the following corrections to the two "data not shown" statements. 

(1) Page 11 lines 253-255 

We have removed the "data not shown" and instead added Figure 4B to Figure 4. The revised parts are shown in red.

Fig 4. Cytokeratin (CK) 13 in the saliva of patients with rheumatoid arthritis (RA) and healthy participants. Twelve microliters of saliva per lane were subjected to sodium dodecyl sulfate-polyacrylamide gel electrophoresis. A: Immunostaining was performed using an anti-CK13 antibody as the primary antibody and horseradish peroxidase-conjugated goat anti-rabbit IgG antibody as the secondary antibody. Lane P: positive control (recombinant CK13), Lanes 1–5: saliva from patients with RA; Lanes 6–11: saliva from healthy participants. B: Immunostaining was performed using only HRPO-conjugated goat anti-rabbit IgG antibody without primary antibody. Lane 1: Saliva from a healthy individual.

(2) Pages 11-12 lines 263-265 

We have removed the "data not shown" and instead added Figure 5B to Figure 5. The revised parts are shown in red.

Fig 5. Citrullinated peptides in the saliva of patients with rheumatoid arthritis (RA) and healthy participants. Twelve microliters of saliva per lane were subjected to sodium dodecyl sulfate-polyacrylamide gel electrophoresis. A: Immunostaining was performed using an anti-modified citrulline antibody as the primary antibody and a horseradish peroxidase-labeled goat anti-rabbit IgG antibody as the secondary antibody after the chemical treatment of the membrane. Lane P: positive control (citrullinated thrombomodulin); Lanes 1–5: saliva from patients with RA; Lanes 6–11: saliva from healthy participants. B: Immunostaining was performed using only HRPO-conjugated goat anti-rabbit IgG antibody without primary antibody. Lane 1: Saliva from a healthy individual.

To Reviewer 1

We appreciate the reviewer's valuable and insightful comments. The added/revised sentences are shown in red in the manuscript.

1. Page 4 line 92. Please give a brief description of the chemical treatment and why it is necessary to perform the chemical treatment on the PVDF membrane. Perhaps the target protein binds to PVDF membrane only if the membrane is chemically treated. Perhaps the antibody binds to citrullinated protein only if the membrane is chemically treated. It is not clear whether the chemical treatment affects the PVDF membrane or proteins on the membrane.

Thank you for your valuable comments. As you pointed out, we also think that the explanation of the identification method of citrullinated peptides was insufficient. In the study of citrullinated protein peptides, detection by AMC (anti-modified citrulline) antibody has been used as the gold standard for the identification of citrullinated peptides since Senshu's report in 1992 (Senshu T et al. Anal Biochem 203, 94, 1992). In this detection method, the citrulline residue of the citrullinated peptide blotted onto PVDF is chemically treated and detected by anti-AMC antibody, instead of chemically treating PVDF to bind the citrullinated peptide. The chemical modification of the citrulline residue is based on a chemical reaction used in the colorimetric quantification of citrulline as a free amino acid.

To describe the specific method and reason for the chemical modification method used to identify citrullinated peptides, the text in Pages 4-5, lines 93-100 has been revised as follows.

(Before correction): This antibody is absolutely specific for citrullinated peptides. In brief, the blotted polyvinylidene difluoride membrane was chemically modified according to the manufacturer’s instructions.

(After correction): This AMC antibody is the gold standard reagent for the detection of citrullinated peptides and is a polyclonal antibody that specifically recognizes chemically modified peptidyl-citrulline residue [22]. The detection method is briefly shown below. To chemically modify the citrulline residues of citrullinated peptides transferred onto a polyvinylidene difluoride (PVDF) membrane, the membrane was chemically modified as per the manufacturer's instructions (ROI004, SHIMA Laboratories) with the following solution: 0.0125% FeCl3, 2.3 M H2SO4, 1.5 M H3PO4, 0.25% diacetyl monoxime, 0.125% antipyrine, and 0.25% acetic acid at 37°C for overnight incubation [22]. 

2. Page 7 line 152. It seems the chemical treatment was performed on the blot after the protein had been transferred from the gel to the PVDF membrane. Does chemical treatment convert arginine to citrulline?

As you mentioned, we performed a chemical treatment on the PVDF membrane to which the peptide was transferred from the gel. This chemical reaction is against the citrulline residue in the peptide on the PVDF membrane and does not convert arginine to citrulline.

3. Page 4 line 92. Please name the manufacturer whose instructions you followed for chemically treating the PVDF membrane.

Following your instruction, I added the company name (SHIMA Laboratories) in red on Page 5 line 98.

4. Page 5 line 117. Please include the name of the manufacturer and the model of the MALDI-TOF mass spectrometer.

As instructed, the name of the manufacturer (Bruker Daltonics) and the model name of the MALDI-TOF mass spectrometer (Microflex LRF20) were added in red in Page 6 lines 130-131.

5. Page 5 line 118. Peptide mass fingerprinting requires the protein to be digested with trypsin. Please describe how the sample was prepared for MALDI-TOF mass spectrometry. Was the protein reduced and alkylated? Were the peptides desalted before application to the target plate? Please describe how the sample was prepared for MALDI-TOF mass spectrometry.

As per your instructions, we have inserted the following text in Page 6, lines 127-

129 regarding the treatment of samples prior to submission to mass spectrometry (trypsin digestion, with or without reduction/alkylation/desalting, and matrix for ionization).

and digested with trypsin (Promega, Madison, WI) without alkylation. The sample solution was desalted using ZipTip (Millipore) and then ionized using �-cyano-4-hydroxycinnamic acid as a matrix [23], and 

6. page 7 lines 151-153. Typing error "one gel was immunostained with the anti-CK13 antibody, and the other gel was chemically treated and immunostained with the AMC antibody. The typing error is the word "gel". The word gel should be blot.

Thank you for pointing out the error. As you indicated, we have changed the word "gel" to "blot" for Page 7 lines 165-166.

(After correction) "One blot was immunostained with the anti-CK13 antibody, and the other blot was chemically treated and immunostained with the AMC antibody." 

7. Page 8 line 189. the UniProt accession number for human cytokeratin 13 protein is P13646. the K1C13 name is the name of the gene.

Thank you for your suggestion. As you pointed out, the UniProt accession number for the human cytokeratin 13 protein was P13646. In Results, in Page 9 line 207 "K1C13_HUMAN" was deleted and "P13646" was added in red.　

8. Page 9 line 207. Typing error FigIP should be IP.

Thank you for pointing out the typographical error. In Results, Page 10 line 226 "FigIP" was corrected to "IP" in red.

9. Please provide the amino acid sequences of cytokeratin 13 peptides by MALDI-TOF mass spectrometry.

Following your instruction, the amino acid sequence of cytokeratin 13, including the amino acid sequence of cytokeratin 13 peptide identified by MALDI-TOF mass spectrometry (underlined), is shown in Supplementary table 2. 

Supplementary table 2. Amino acid sequences of cytokeratin 13 peptide identified by matrix 2 assisted laser desorption/ionization time of flight (MALDI-TOF) mass spectrometry.

1 MSLRLQSSSA SYGGGFGGGS CQLGGGRGVS TCSTRFVSGG SAGGYGGGVS

51 CGFGGGAGSG FGGGYGGGLG GGYGGGLGGG FGGGFAGGFV DFGACDGGLL

101 TGNEKITMQN LNDRLASYLE KVRALEEANA DLEVKIRDWH LKQSPASPER

151 DYSPYYKTIE ELRDKILTAT IENNRVILEI DNARLAADDF RLKYENELAL

201 RQSVEADING LRRVLDELTL SKTDLEMQIE SLNEELAYMK KNHEEEMKEF

251 SNQVVGQVNV EMDATPGIDL TRVLAEMREQ YEAMAERNRR DAEEWFHTKS

301 AELNKEVSTN TAMIQTSKTE ITELRRTLQG LEIELQSQLS MKAGLENTVA

351 ETECRYALQL QQIQGLISSI EAQLSELRSE MECQNQEYKM LLDIKTRLEQ

401 EIATYRSLLE GQDAKMIGFP SSAGSVSPRS TSVTTTSSAS VTTTSNASGR

451 RTSDVRRP 

Healthy human saliva was separated by two-dimensional electrophoresis, and the part of the gel that matched the spot of citrullinated peptide was cut and analyzed by MALDI-TOF mass spectrometer (Microflex LRF20 [Bruker Daltonics]) and mass spectrometry using peptide mass fingerprinting (Genomine). The underlined part is the amino acid sequence matched to the cytokeratin 13 peptide.

To Reviewer 2

We appreciate the reviewer's valuable and insightful comments. The added/revised sentences are shown in red in the manuscript.

1. The authors should explain why CK13 was focused on as a protein in saliva.

Thank you for pointing this out. We apologize for the confusing description. This study did not hypothesize that CK13 is a peptide in saliva, but was a comprehensive search for citrullinated peptides in saliva. We are discussing CK13 because our analysis showed that CK13 is present in saliva. To avoid misunderstanding, we added the following words (in red) to Page 3 lines 71-72.

“This study aimed to identify previously unidentified citrullinated peptides in human saliva through a comprehensive investigation of human saliva and to discuss their clinical significance.”

2. it is unclear how serum collected from RA and healthy individuals were used in this study.

Thank you for pointing this out. Serum was not used in this study, and it was a typographical error; we have removed "and serum" from Page 4 lines 76-77.

(Before correction): saliva and serum samples were collected from participants 

(After correction): saliva samples were collected from participants 

3. The patients' characteristics, including smoking history and serological profile of healthy subjects and RA patients, should be appropriately described.

Following your instructions, we made a table on the characteristics of the healthy subjects and patients with RA whose saliva were collected. We show this as Supplementary table 1. For healthy subjects, we did not present serological characteristics because they did not suffer from RA, and only smoking history was described. For patients with RA, serological profiles were described.

Supplementary table 1. Characteristics of healthy subjects (A, B) and rheumatoid arthritis (RA) patients (C) whose saliva were collected.

A

Healthy individual Age (years) Sex Smoking history

1 31 Male (-)

2 50 Male (+)

3 44 Male (-)

4 37 Male (-)

5 34 Male (-)

6 33 Male (-)

7 28 Male (-)

8 27 Male (-)

9 27 Male (-)

10 27 Male (-)

B

Healthy individual Age (years) Sex smoking history

1 31 Male (-)

2 29 Female (-)

3 29 Male (-)

4 30 Male (-)

5 32 Male (-)

6 32 Female (-)

7 31 Female (-)

8 29 Female (-)

9 31 Female (-)

10 29 Female (-)

C

RA

 Age (years) Sex CRP (mg/dl) RF

(IU/ml) MMP-3

(ng/ml) Anti-CCP antibody (U/ml) Smoking history

1 85 Female 1.1 ＜3.0 385.8 15.6 (-)

2 65 Female 0.04 54 66.6 unknown (-)

3 66 Female 0.19 895.6 69.8 3.6 (+)

4 58 Female 0.13 28.3 99.1 72.7 (-)

5 64 Male 0.38 73.3 170.4 73.1 (+)

6 70 Female 0.06 72.8 30.9 290 (+)

7 88 Male 0.26 6.3 270.3 12.8 (+)

8 49 Female 1.79 173.6 297.6 243 (-)

9 69 Female 0.03 22.3 58.6 1280 (-)

10 55 Female 0.34 106.2 97.6 659 (+)

11 50 Female 0.02 42.3 40.4 52.9 (-)

　

A: Healthy individuals used for detection of citrullinated peptides in saliva; B: Healthy individuals used for semi-quantification of Citrullinated Cytokeratin13 (Cit-CK13) in saliva; C: RA patients used for semi-quantification of Cit-CK13 in saliva

Smoking history: including current smoking. CRP: C-reactive protein (normal range 0.3 

mg/dL or less); RF: Rheumatoid Factor (normal range 15 IU/mL or less); MMP-3: Matrix 

Metalloproteinase-3 (normal range: female 17.3-59.7 ng/mL, male 36.9-121 ng/mL)

4. the authors should explain how 10 patients to be compared with RA patients were selected from 21 healthy controls.

There was a counting error when we wrote the number of saliva samples collected. We collected saliva samples from 20 healthy subjects in total, 10 for the Western blot experiment to detect citrullinated peptide in saliva and 10 for the semi-quantitative study of cit-CK13 in saliva.

Page 1, line 21.

(Before correction): Saliva samples were collected from 11 patients with RA and 21 healthy individuals.

(After correction): Saliva samples were collected from 11 patients with RA and from 20 healthy individuals.

Page 4, lines 78-81.

(Inserted text): For saliva from healthy individuals, samples were used for experiments in the order of collection, and 10 subjects were used for experiments to detect citrullinated peptides in saliva by Western blot, and the other 10 subjects for semi-quantitative study of the citrullinated peptide identified in saliva by mass spectrometry. 

(Before correction): Saliva and serum samples were collected from subjects with no subjective symptoms of periodontal disease (11 rheumatoid arthritis patients and 21 healthy individuals) after screening using the questionnaire of Sugihara et al [21].

(After correction): Saliva samples were collected from subjects with no subjective symptoms of periodontal disease (11 rheumatoid arthritis patients and 20 healthy individuals) after screening using the questionnaire of Sugihara et al [21].

Page 8, lines 186-188.

(Before correction): The participants had a median age of 31.0 years (95% confidence interval: 29.5–34.7; 15 males, six females). 

(After correction): The participants had a median age of 32.0 years (95% confidence interval: 28.1–39.4; 10 males). Supplementary table 1A shows the age and smoking history of the 10 participants.

5. "p = 0.705" in the 242nd line should be corrected to "p = 0.605" as shown in Fig 

Thank you for pointing out the error. We have corrected the description of Figure 5 in Page 12 line 270 as shown below.

"As shown in Fig 6, there was no statistically significant difference between the salivary Cit-CK13 ratios of patients with RA (n = 11) and healthy individuals (n = 10; p = 0.605)."

---

## [Editor Report · Decision Letter 1]

7 Mar 2022

Detection of salivary citrullinated cytokeratin 13 in healthy individuals and patients with rheumatoid arthritis by proteomics analysis

PONE-D-21-37361R1

Dear Dr. Yasuda,

Thank you for modifying your manuscript according to suggestions from the reviewers. We’re pleased to inform you that your manuscript has been judged scientifically suitable for publication and will be formally accepted for publication once it meets all outstanding technical requirements.

Kind regards,

Oksana Lockridge, Ph.D.

Academic Editor

PLOS ONE

---

## [Editor Report · Acceptance letter]

11 Mar 2022

PONE-D-21-37361R1 

Detection of salivary citrullinated cytokeratin 13 in healthy individuals and patients with rheumatoid arthritis by proteomics analysis 

Dear Dr. Yasuda:

I'm pleased to inform you that your manuscript has been deemed suitable for publication in PLOS ONE. Congratulations! Your manuscript is now with our production department. 

Kind regards, 

on behalf of

Dr. Oksana Lockridge 

Academic Editor

PLOS ONE